# Systematic comparison of Mendelian randomisation studies and randomised controlled trials using electronic databases

Maria K Sobczyk ,[1] Jie Zheng ,[1,2,3] George Davey Smith ,[1] Tom R Gaunt [1]

## ABSTRACT

**Objective** To scope the potential for (semi)-automated triangulation of Mendelian randomisation (MR) and randomised controlled trials (RCTs) evidence since the two methods have distinct assumptions that make comparisons between their results invaluable.

**Methods** We mined ClinicalTrials.Gov, PubMed and EpigraphDB databases and carried out a series of 26 manual literature comparisons among 54 MR and 77 RCT publications.

**Results** We found that only 13% of completed RCTs identified in ClinicalTrials.Gov submitted their results to the database. Similarly low coverage was revealed for Semantic Medline (SemMedDB) semantic triples derived from MR and RCT publications –36% and 12%, respectively. Among intervention types that can be mimicked by MR, only trials of pharmaceutical interventions could be automatically matched to MR results due to insufficient annotation with Medical Subject Headings ontology. A manual survey of the literature highlighted the potential for triangulation across a number of exposure/outcome pairs if these challenges can be addressed.

**Conclusions** We conclude that careful triangulation of MR with RCT evidence should involve consideration of similarity of phenotypes across study designs, intervention intensity and duration, study population demography and health status, comparator group, intervention goal and quality of evidence.

## STRENGTHS AND LIMITATIONS OF THIS STUDY

⇒ Mendelian randomisation (MR) has become a popular method in causal inference in genetic epidemiology, and while often used as proxy for to randomised clinical trials (RCTs), little is known about scope for automatic comparison between MR and RCT results.

⇒ Previous research has established conceptual similarities and differences between MR and RCT methodology, however, without focus on applied cases.

⇒ The study found that a low percentage of completed RCTs were submitted to ClinicalTrials.gov and that a similarly low coverage was found for MR and RCT publications in Semantic Medline. Only trials of pharmaceutical interventions could be automatically matched to MR results due to insufficient annotation with Medical Subject Headings ontology among other interventions.

⇒ Following manual extraction of MR and RCT literature, we assessed result concordance across the two methods and discussed multiple possible reasons for discrepancies.

⇒ Sparsity of data in electronic databases hinders the ability to automatically compare results of MR and RCT studies. In the absence of retrospective manual extraction of MR and RCT results from publications, more research effort needs to be spend developing machine-learning approaches to aid systematic comparisons. Our study helps identify study design features which need to be captured by such methods.

## BACKGROUND

Randomised controlled trials (RCTs) are deemed the 'gold standard' in evaluating the efficacy of interventions and guiding practice in clinical research, with well-established methodology.[1] In RCTs, a selection of individuals intended to represent the target population is randomly assigned to a treatment or control group, allowing estimation of the intervention's effectiveness in the absence of confounding variables and reverse causality that are present in observational studies. In the past two decades, an approach to causal inference using natural genetic variation, known as Mendelian randomisation (MR)— usually implemented as an instrumental variable (IV) analyses—has gained popularity.[2,3] This approach has been referred to as 'nature's randomised trials'[4] and is based on the randomisation from parents to offspring of genetic variants encapsulated in Mendel's laws of segregation and independent assortment.[2,5] At a population level, the randomisation is approximate, but still allows genetic variants that are robustly associated with the measured exposure to be used to estimate

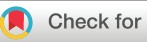

For numbered affiliations see end of article.

**Correspondence to**
Dr Maria K Sobczyk;
maria.sobczyk@bristol.ac.uk

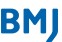

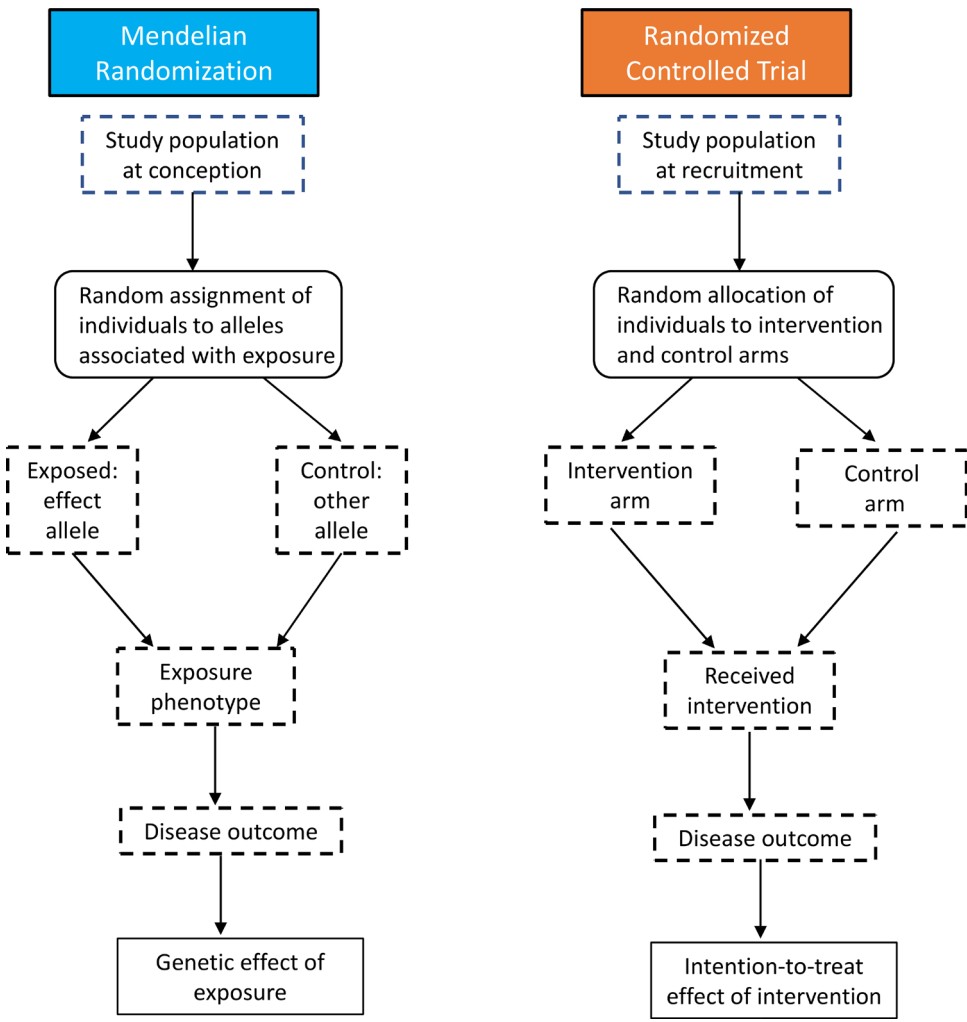

**Figure 1** Comparison of Mendelian randomisation and randomised controlled trial design. After: Nitsch *et al*,[171] Ebrahim and Smith[172] and Ference.[157]

the unbiased causal effect of an exposure (generally acting across life) on health outcomes, as long as certain assumptions, discussed in detail elsewhere,[2 3 6] are met.

Despite drawing on observational data, the MR approach broadly aligns with that of an RCT, where the goal is to estimate the causal effect of an intervention on the given endpoint based on groups (arms) which do not differ with respect to confounding variables (figure 1). However, since in MR randomisation takes place at conception, the time lag to the start of outcome recoding is longer[7] compared with RCTs, where median duration of phase 3 trials is 40 months.[8] Similarly to RCTs, most MR analyses should be free of confounding and reverse causation bias due to variants being allocated randomly before birth and outcome condition onset.

Previous research has shown examples of evidence triangulation where MR results predicted the overall RCT results based on totally orthogonal data with an unrelated set of systematic errors and biases.[9] For instance, MR demonstrated the lack of effect of genetically predicted concentrations of High-density lipoprotein cholesterol (HDL-C) on cardiovascular events[10–12] as well as selenium in prostate cancer prevention trials.[13 14] On the other

hand, MR showed the beneficial effect of lifelong endogenous low-density lipoprotein cholesterol (LDL-C) levels,[15] HMG-CoA reductase inhibition (statin drug target) and PCSK9 inhibition on cardiovascular disease (CVD),[15 16] while predicting also the increased risk of type 2 diabetes (T2D) as a side effect of statin usage. However, three independent MR studies were at odds with later RCTs by predicting increased risk of T2D also as a side effect of PCK9 inhibition.[17]

There are several possible explanations for apparent or real discordance in the results of RCT and MR studies. These range from different durations, magnitude and time-varying nature of the exposure, origin of the study populations, and natural genetic variation imperfectly mimicking the molecular action of the drug, some of which we explore. The direct comparison of MR and RCT findings is facilitated by the use of a precisely defined estimand,[18] for example, the effect on incident coronary heart disease risk of lowering LDL cholesterol by 1 mmol/L for 5 years. While RCTs will estimate something close to this, and be scalable to it, with MR studies the exposure difference associated with the genetic instruments will often exist from birth (or before) and

may change in magnitude over time.[19] This is discussed further in online supplemental file 1.

While RCTs can provide the highest-quality evidence, they may have limitations. They are often expensive to carry out, can be of small size and lack external validity,[20] have short follow-up and typically take place after disease onset.[21 22] As in other study types, RCT results may be flawed due to poor design and execution, for example, imperfect randomisation, unblinding and differential loss to follow-up between study arms.

Unlike RCTs, MR studies are inexpensive and quick to perform when suitable genetic instruments are available. Therefore, they can potentially prioritise intervention–condition pairs to assess in RCTs. Moreover, it has been proposed that MR also guides the design of RCTs, improving eligibility criteria to prioritise groups most likely to benefit, suggesting diseases for composite endpoint construction and alerting to potential side effects.[1 23] Since MR analyses suffer a different set of biases than RCTs, MR evidence can be used to complement RCTs and other study designs in the triangulation framework to guide therapeutic development and clinical practice.[24–26] Finally, the extensive use of existing observational data for MR enables intervention targets to be evaluated in a wider range of subpopulations than is feasible for RCTs (improving generalisability), and allows comparisons to be made that might be unethical in experimental studies, for example, when there is strong evidence in favour of a particular treatment.

The goal of this research is to survey the extent of concordance between MR and RCT studies to date and identify possible factors for disparities in the direction of effect, which limit the ability to extrapolate from MR results to RCTs and increase the complexity of the triangulation process. In this study, we aimed to carry out a systematic analysis of MR and RCT results using automated mining of data in the public domain, including the ClinicalTrials.Gov,[27] EpigraphDB[28] and PubMed databases. We evaluate the comprehensiveness and scope of the data available and potential for comparative analyses between MR and RCTs. We then go on to develop a series of case studies looking in detail at MR and RCT comparisons across 26 exposure–outcome pairs. Throughout, we use the term 'intervention' as synonymous with 'exposure' and 'condition' as synonymous with 'outcome'.

## METHODS

### ClinicalTrials.Gov data sources
All ClinicalTrials.Gov study data are available for download as PostgreSQL database from the database for Aggregate Analysis of ClinicalTrials.gov (AACT)[29] released by the Clinical Trials Transformation Initiative.[30] We downloaded its static release from 1 July 2023. Processing of the database files was carried out using custom Python and R scripts.

### ClinicalTrials.Gov data filtering
We filtered the ClinicalTrials.Gov studies using a number of criteria to identify RCTs with submitted results allowing direct comparison with MR studies. These are depicted in figure 2A and provided in detail in online supplemental note.

### EpigraphDB queries
EpigraphDB[28] was used to collect information about confirmed drug-target associations which were initially sourced from the Open Targets Platform[31] and verified in DrugBank.[32] EpigraphDB was then used to retrieve expression quantitative trait loci (eQTL) and protein quantitative trait loci (pQTL) MR results previously described in Zheng *et al*.[33] We also used EpigraphDB[28] to source SemMedDB[34] V.1.8 semantic triples associated with select MR and RCT publications identified by PubMed. SemMedDB triples in EpigraphDB are prefiltered for annotation of epidemiological studies as described previously.[35]

### PubMed data harvesting
We searched PubMed for all RCT and MR studies published before 2023 on 1 July 2023. For RCTs, we searched titles and abstracts for keywords: "randomized controlled trial" or "RCT" and we used PubMed's in-built *Randomized Controlled Trial* label filter to obtain more specific hits, reducing the number of hits from 129 077 to 74 559. In order to retrieve potential MR studies, we used the keywords: "mendelian randomization" or "mendelian randomisation". We also considered using a Medical Subject Headings (MeSH) label "Mendelian Randomization Analysis" but it returned an unrealistically low number of hits (3174), a consequence of manual indexing.

### Literature searches
We used Semantic Scholar and Google Scholar to survey MR and RCT literature indexed before 1 July 2023. We queried the databases with the following search terms: "[exposure] [condition] Mendelian Randomization" and "[exposure] [condition] Randomized Controlled Trial". The articles were initially screened by title and abstract. We considered original research MR, RCT studies as well as meta-analyses. We included 26 intervention–outcome pairs to represent a wide array of behavioural and nutritional interventions with a diverse set of common disease (cardiometabolic, neuropsychiatric, cancer, dermatological) and disease biomarker outcomes, based on our expert knowledge of the field. Our chosen exposures correspond to the top four modifiable risk factors accounting for 39% of deaths in the USA[36]: high alcohol intake, high body mass index (BMI), lack of exercise and smoking. In addition, since potentially preventative effects of nutritional factors are controversial and notoriously difficult to evaluate using non-randomised study designs,[37] we also included vitamins D and E as well as coffee as an intervention. We acknowledge that this

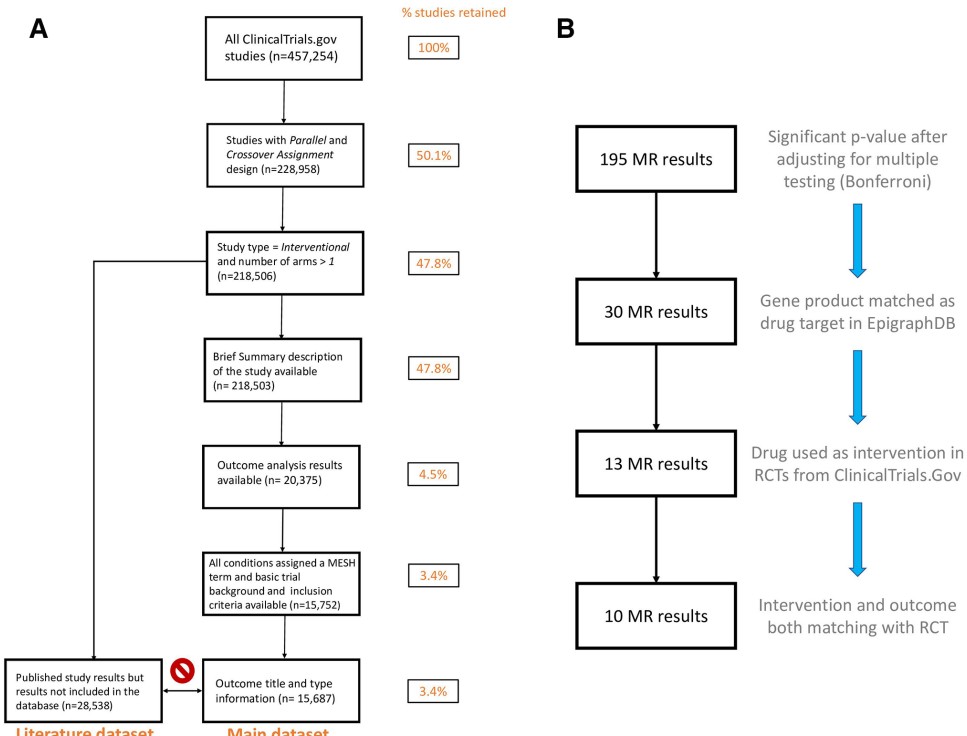

**Figure 2** (A) Filtering steps applied to ClinicalTrials.Gov database. Filtering was designed to identify RCTs whose final results statistics were uploaded to the database (main dataset). In addition, other RCTs which published their findings in scientific journals were identified (literature dataset). (B) Filtering steps applied to EpigraphDB database. Filtering was designed to identify protein QTL MR studies with intervention and exposure matching those of RCT published on ClinicalTrials.Gov. MR, Mendelian randomisation; QTL, quantitative trait loci; RCT, randomised controlled trial.

choice is somewhat subjective but we believe it to be illustrative of the current MR and RCT literature.

MR and RCT studies were compared across: population characteristics (sex, ethnicity, age, health status), comparator group, goal of intervention (prevention or treatment/slowing progression), direction of effect, length of follow-up, main test statistic in the study and its impact as judged by citation number.

### Patient and public involvement

Patients or the public were not involved in the design, or conduct, or reporting, or dissemination plans of our research.

### RESULTS
### ClinicalTrials.Gov data overview

In total, we found 457 254 individual studies were registered with a unique ClinicalTrials.gov identifier. We filtered them using a number of steps to identify RCTs and facilitate comparison with MR (figure 2A). In our analysis, we identified 218 506 RCT studies (48% of the total). To allow semiautomated comparison with MR studies, we focused on the study subset which submitted their statistical analysis results to the database (referred to as the *main dataset*, online supplemental dataset 1). However, we found that only 3.4% of studies—15 752 met this criterion, along with including background information on the trial. To expand that number, we also considered an

additional 28 538 RCT studies which did not publish their results in ClinicalTrials.gov but instead linked to a peer-reviewed publication (referred to as the *literature dataset*, online supplemental dataset 2).

The majority of RCTs in the main dataset followed parallel assignment of participants to treatment (online supplemental figure S1a), most were designed for treatment (n=12 336, online supplemental figure S1b), rather than prevention (n=1613) and the vast majority of them had been completed (online supplemental figure S1c). More trials were observed to be in phase 3 than 4 (online supplemental figure S1d), most trials included both males and females (online supplemental figure S1e) and a great majority had two arms (online supplemental figure S1f). The median number of primary outcomes was 1 (online supplemental figure S2a), with a median of 6 secondary outcomes (online supplemental figure S2b). Over half of studies report at least one result with p value less than 0.05 (online supplemental figure S2c). Comparison with features of all RCTs in the database showed that our selection was broadly representative (online supplemental dataset 3), although our dataset was enriched for completed and late-phase trials.

### Suitability of MeSH annotation

In order to attempt automated matching of RCTs and MRs involving similar interventions and outcomes for RCTs and MR, we needed to first establish the quality

**Table 1** Completeness of MeSH term annotation among the chosen intervention types in the *main* (RCT results available in ClinicalTrials.Gov) and *literature* (RCT results unavailable in ClinicalTrials.Gov but study linked to a publication with results) datasets

| Dataset | Intervention type | Total | Intervention MeSH missing | % missing | Condition MeSH missing | % missing |
|---|---|---|---|---|---|---|
| Main | Drug | 11 537 | 2212 | 19.2 | 992 | 8.6 |
| Literature | Drug | 10 927 | 1773 | 16.2 | 1031 | 9.4 |
| Main | Behavioural | 1239 | 1075 | 86.8 | 242 | 19.5 |
| Literature | Behavioural | 3017 | 2815 | 93.3 | 620 | 20.5 |
| Main | Dietary supplement | 242 | 116 | 47.9 | 31 | 12.8 |
| Literature | Dietary supplement | 830 | 581 | 70.0 | 164 | 19.8 |

MeSH, Medical Subject Headings; RCT, randomised controlled trial.

of annotation of RCTs with MeSH in ClinicalTrials.Gov. The most common intervention was drug (online supplemental table S1). Since we were only interested in the intervention types which can be instrumented by MR, we also focused on the fifth and seventh most popular types of interventions: behavioural and dietary supplement. We found that MeSH intervention annotations were missing for only 19% and 16% of drug interventions in the main and literature datasets, accordingly (table 1). However, the overwhelming majority of RCTs in the behavioural and dietary supplement category did not contain a MeSH intervention term. Due to well-standardised disease taxonomy, a much lower level of missing data was found for MeSH condition terms. This allowed us to proceed with automated analysis of drug RCT data; however, for behavioural and dietary supplement we were only able to do a manual screening for RCTs with corresponding MR studies.

### Pharmaceutical interventions in RCTs and MR

Genetic IVs in MR can be used as proxies for pharmaceutical interventions in RCTs. pQTL or eQTL, that is, variants associated with expression of protein drug targets are used to directly proxy the action of a drug. Here, we use the biggest MR dataset for drug target protein–disease associations, examined in whole blood, from Zheng *et al*.[33] We focused on *cis*-acting instruments as a more specific marker for drug efficacy as *trans*-instruments are more likely to be pleiotropic, potentially leading to spurious results.[23]

We matched the drug target proteins in Zheng *et al*[33] with drug–gene associations sourced from EpigraphDB[28] (figure 2B). This allowed us to merge the Zheng *et al*[33] dataset with the main and literature RCT dataset via the drug listed in EpigraphDB and MeSH drug intervention term, accordingly. For the outcome, we were then able to match RCTs and MR manually due to the reasonably

**Table 2** Drug target–disease matches supported by evidence from MR (blood pQTL instruments[33]) and RCT studies (*main dataset* from ClinicalTrials.Gov)

| MR exposure | RCT drug intervention | MR outcomes | RCT conditions | Matching trials | Concordant direction of effect? | xQTL |
|---|---|---|---|---|---|---|
| PCSK9 | Evolocumab, alirocumab | Non-cancer illness code self-reported: high cholesterol \|\| id:UKB-a:108 | Hyperlipidaemia, dyslipidaemia, hypercholesterolaemia, mixed dyslipidaemia | 25 | Yes | pQTL |
| APOB | Mipomersen | LDL cholesterol \|\| id:300, HDL cholesterol \|\| id:299, triglycerides \|\| id:302, non-cancer illness code self-reported: high cholesterol \|\| id:UKB-a:108, total cholesterol \|\| id:301, | Hyperlipidaemia, dyslipidaemia, hypercholesterolaemia, mixed dyslipidaemia | 6 | Yes | pQTL |
| IL12B | Ustekinumab | Non-cancer illness code self-reported: psoriasis \|\| id:UKB-a:100; ulcerative colitis \|\| id:970; Crohn's disease \|\| id:12; inflammatory bowel disease \|\| id:294 | Psoriasis, psoriatic arthritis, Crohn's disease, colitis, inflammatory bowel disease | 21 | Yes | pQTL |

MR, Mendelian randomisation; pQTL, protein quantitative trait loci; QTL, quantitative trait loci; RCT, randomised controlled trial.

low number of hits. The results displayed in table 2 show overlap of the RCT and MR datasets. We found four drugs: evolocumab/alirocumab, ustekinumab and mipomersen that share support from both MR and RCT studies. Evolocumab/alirocumab and mipomersen inhibit key players (PCSK9 and apoB) in lipid transport helping to lower plasma LDL-C levels.[38] The Zheng *et al*[33] MR study showed a negative effect of reduced PCSK9 levels on high cholesterol in the UK Biobank, while in 25 RCT studies drug-induced abrogation of PCSK9 activity led to positive outcomes in the treatment of hyperlipidaemia, hypercholesterolaemia and dyslipidaemias in general. Similarly, reduced expression/activity of apoB in MR and six RCT studies resulted in genetically predicted lower levels of LDL cholesterol and total cholesterol in the UK Biobank as well as improved outcomes in the treatment of dyslipidaemias, respectively. The third example of a good match between RCT and MR studies concerns inhibition of the p40 subunit of interleukin 12 and 23 (IL12B).[39] Both MR and 21 RCTs show benefit of inhibition of p40 on immune-mediated disease: psoriasis and inflammatory bowel disease.

In general, pQTL MR-based prediction of drug target–condition pairs offered good recall when compared with the pairs in the Open Targets Platform for the proteins with MR evidence. The only drug target indications missing included conditions not analysed in the MR study, such as CD33 protein being the drug target for treatment of leukaemia, with the exception of acetylcholinesterase whose inhibitors (galantamine, donepezil, rivastigmine) are used for treatment of cognitive decline in Alzheimer's disease.[40]

We also compared the RCT dataset with Zheng *et al* blood transcript expression (eQTL)-derived MR analysis (available in EpigraphDB: https://epigraphdb.org/xqtl, online supplemental figure S3). In total, we identified 15 drug target–disease matches in the eQTL dataset (online supplemental table S2), although unlike in the pQTL matches, the direction of effect in MR was incorrect in eight cases. Nevertheless, the eQTL MR results agreed with some well-known drug effects: HDL-C and LDL-C lowering action of CETP and HMGCR inhibitors,[41] respectively, and blood pressure-lowering action of ACE inhibitors.[42]

### PubMed-sourced MR and RCT studies
In addition to searching through the ClincalTrials.Gov database, we also queried PubMed for RCT and MR publications. In total, we found 5135 MR studies published since mid-2000s and 73 306 RCTs published since 1970 until 2022 (figure 3).

### Semantic analysis with SemMedDB
We subsequently wanted to establish the thematic overlap between MR and RCT studies using an alternative method involving semantic analysis. SemMedDB[34] provides a vast repository of semantic predications (subject–predicate–object triple, for example, LDL-C causes ischaemic

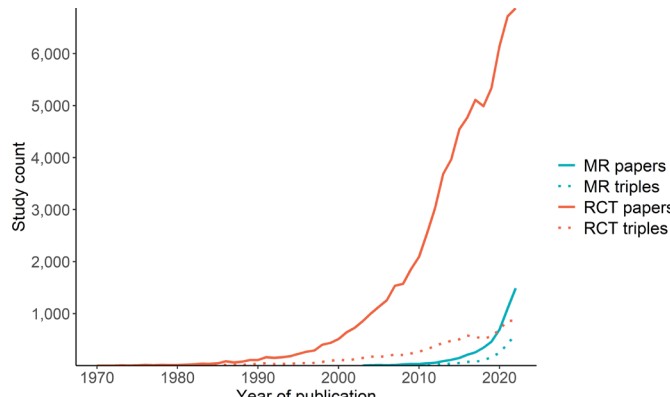

**Figure 3** Popularity of MR and RCT studies over time. We compare counts of MR and RCT papers indexed by PubMed (solid lines) with number of semantic triples derived from them using SemMedDB (dashed lines). MR, Mendelian randomisation; RCT, randomised controlled trial.

heart disease). We linked the MR and RCT publications identified by our PubMed search to their corresponding SemMed triples in EpigraphDB using PubMed ID (online supplemental dataset 4). Overall, only 12% and 36% of RCT and MR papers, respectively, had a semantic triple associated with them (figure 3). When ignoring the predicate, and focusing only on the subject and object, we found a total of 15 113 unique exposure–outcome pairs (online supplemental figure S4), discussed in detail in the online supplemental note. However, only 221 of these were found to be shared across MR and RCT studies.

We then investigated the 221 matching subject–object pairs between MR and RCT studies (online supplemental table S3), as well as individual top counts among subjects (online supplemental table S4) and objects (online supplemental table S5). T2D, insulin and obesity were found among the top shared risk factors, along with lipids and vitamin D. Top outcomes included T2D, CVD, COVID-19 and Alzheimer's disease.

### Case studies of matching MR and RCTs
Since our semiautomatic mining of MR and RCT literature brought limited results for behavioural and nutritional interventions, we selected 26 intervention–outcome case studies by manual mining of the literature representing common lifestyle risk factors, dietary and behavioural exposures, paired with common cardiovascular, glycaemic, neuropsychiatric, musculoskeletal, autoimmune and cancer outcome phenotypes. In total, we surveyed 54 MR and 77 RCT publications (RCTs and meta-analysis of RCTs, online supplemental dataset 5, figure 4) which were systematically compared across several criteria shown in sample online supplemental table S6, and encompass those in the popular PICO (Population, Intervention, Comparison and Outcome) framework.[43]

There, we compare an MR study and two RCT meta-analyses on the effect of vitamin D supplementation in multiple sclerosis (MS).[44–46] While the RCTs looked at

| exposure | outcome | MR | RCT |
|---|---|---|---|
| alcohol | hypertension | 2 | 4 |
| blood pressure | cardiovascular disease | 3 | 3 |
| body mass index | cardiovascular disease | 2 | 2 |
| body mass index | hypertension | 4 | 3 |
| body mass index | type 2 diabetes | 5 | 3 |
| coffee | glycemic biomarkers | 1 | 2 |
| coffee | lipids | 3 | 3 |
| diary intake | blood pressure | 1 | 1 |
| exercise | bone mineral density | 1 | 2 |
| exercise | depression | 2 | 7 |
| exercise | glycemic biomarkers | 1 | 2 |
| exercise | lipids | 1 | 6 |
| exercise | pain | 1 | 1 |
| exercise | schizophrenia | 2 | 3 |
| smoking | lipids | 2 | 2 |
| vitamin D | atopic dermatitis | 2 | 4 |
| vitamin D | blood pressure | 3 | 2 |
| vitamin D | body mass index | 2 | 2 |
| vitamin D | bone fractures | 1 | 5 |
| vitamin D | bone mineral density | 3 | 2 |
| vitamin D | cardiovascular disease | 1 | 2 |
| vitamin D | depression | 4 | 3 |
| vitamin D | glycemic biomarkers | 1 | 3 |
| vitamin D | lipids | 3 | 4 |
| vitamin D | multiple sclerosis | 1 | 2 |
| vitamin E | prostate cancer | 2 | 4 |

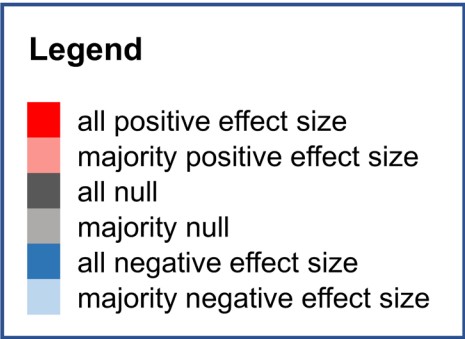

**Legend**

- all positive effect size
- majority positive effect size
- all null
- majority null
- all negative effect size
- majority negative effect size

**Figure 4** Summary of case series of MR and RCT studies with matching exposures (interventions) and outcomes (conditions). The values correspond to the number of analysed studies in a given category, while the cell background colour indicates summary direction of effect on the outcome when exposure is increased—we report direction of effect found either in all analysed studies or their majority (>50%). MR, Mendelian randomisation; RCT, randomised controlled trial.

**Table 3** Overview of discussed criteria for assessment of alignment of MR and RCT study features

| Match criterion | Issues to consider | Example |
|---|---|---|
| Exposure (intervention) | Similarity between analysed exposures | Different types of exercise |
| | Intervention intensity | Vitamin D dosage |
| Intervention goal | Prevention or treatment | Schizophrenia onset or treatment |
| Outcome (condition) | Single or composite outcome | Single or composite cardiovascular outcomes |
| | Binary versus categorical outcome | Depression or rating on depression assessment scale, such as Hamilton Depression Rating Scale |
| | Similarity between analysed outcomes | Different measures of adiposity |
| Source population | Demographics | Young adults or elderly |
| | Health status | Diabetic or healthy |
| Comparator group | Exposure-naïve or previously exposed | Ex-smoker or never-smoker |
| | Active intervention or placebo | Statin as comparator or placebo |
| Duration of intervention/follow-up | Length of intensive intervention and follow-up | Short duration of intervention (<6 months) or long duration and follow-up (>3 years) in RCT and MR |
| | Not uniform intervention intensity or duration | Weekly counselling during the initial phase of the trial or throughout |

MR, Mendelian randomisation; RCT, randomised controlled trial.

potential therapeutic effect of vitamin D in patients with diagnosed MS over 6 months–2 years: measured disability (Expanded disability status scale (EDSS) Score) and recorded relapses as outcomes, the MR study took place in the general population and measured the causal effect of genetically predicted lifetime circulating vitamin D concentrations on prevention of MS. The conclusions of MR and RCT studies did not align well, with MR analyses providing evidence for reduced risk of MS conferred by higher vitamin D levels, but no significant therapeutic effect of vitamin D in existing MS was found in the five small meta-analysed trials. Differences which may impact on the ability for MR to complement RCT studies are summarised in table 3 and discussed below based on this series of case studies.

### Exposure/intervention

We found that overlapping MR and RCT interventions are often not perfectly identical which may impact on the estimated direction of effect. For example, MR exercise exposures are based on genetic variants associated with self-reported physical activity (moderate-to-vigorous and vigorous)[47 48] in studies assessing the effect on both lipids and bone mineral density (BMD). However, the corresponding RCTs used particular types of exercise, such as walking,[49] aerobic exercise,[50 51] progressive resistance training[52] and maximal strength training[53] as interventions. While an MR study[48] and two trials[53 54] showed concordant (figure 4), positive effect of exercise on BMD, we found that the effect of exercise on lipids did not match between MR and RCTs, with MR study[47] reporting null effect and trials generally finding positive effects

on HDL-C concentration and negative on LDL-C, total cholesterol and triglycerides blood levels.[50–52 55]

Furthermore, intensity of intervention can affect the comparative value of MR and RCT study conclusions. The MR study of vitamin D levels on bone fractures[56] was only able to assess linear effects of the normal range of circulating 25-hydroxyvitamin D concentrations. Consequently, the positive effect of high-concentration vitamin D (≥700 IU daily) on bone fractures in the elderly seen in RCTs[57–59] may not have been accessible in the MR study.

### Intervention goal

The intervention goal between MR and RCT studies can match (both prevention or treatment) or be misaligned which can potentially impact the ultimate conclusions of the study. We found the latter to be the case for the effect of exercise on schizophrenia. Two MR studies found a null preventative effect of exercise on schizophrenia,[60 61] while three meta-analyses of RCTs found a consistent effect of a variety of exercise types on improving total and negative symptoms of schizophrenia[62–64] (figure 4).

### Outcome

The short duration of RCTs mean some outcomes (eg, myocardial infarction) do not accumulate enough events to detect a significant effect, therefore composite measures grouping related diseases are often used. When comparing the effect of systolic blood pressure (SBP) on CVD outcomes, we found matching conclusions with elevated SBP increasing the risk of CVD both in RCT[65–67] and MR[68–70] studies (figure 4), with MR studies using both single disease outcomes and a composite outcome.

However, MR studies analysing the impact of BMI on CVD found reduced adiposity led to reduction in arterial hypertension, CVD and stroke,[71 72] which contrasted with the results of one of the biggest RCTs to date. The Look AHEAD RCT in older patients with T2D found no preventative effect of weight loss on a composite outcome relating to mortality from cardiovascular causes, non-fatal myocardial infarction, non-fatal stroke or hospitalisation for angina.[73]

Second, RCT outcomes are often on a quantitative scale measuring symptom strength according to established metrics, (eg, depressive symptoms on Hamilton Depression Rating Scale[74]). However, the best disease Genome-wide association study (GWAS) used to identify MR instruments often represent binary disease outcomes, which could potentially lead to differential conclusions due to reduced power to detect subtler therapeutic effects. While exercise is causally associated with reduced depression and depressive symptoms both in MR[75 76] and RCTs,[62 74 77–80] the differences in outcome phenotypes could potentially contribute to null MR results[81 82] and positive effect of vitamin D on attenuating eczema symptoms in RCTs.[83–86]

### Source population

MR studies are likely to draw from a wider demographic than RCTs due to use of biobanks and GWAS consortia, while RCTs focus on high risk groups.[23] For example, while in the MR study conducted in general population, there was no strong significant effect of exercise on glycaemic markers: Hemoglobin A1c (HbA1c), fasting glucose and Homeostatic model assessment for insulin resistance (HOMA-IR),[87] a significant reduction was found in the meta-analysis of 32 RCTs involving patients with T2D.[88]

On the other hand, five MR [89–93] studies along with three large RCTs[94–96] consistently provide evidence that weight loss is causally associated with reduced risk of T2D (figure 4), despite MR including the general population and RCTs focusing on at-risk individuals with impaired glucose tolerance.

As another example of possible demographics-driven differences in trial and MR results, MR studies on the relationship between vitamin D levels and atopic dermatitis were conducted in the general population,[81 82] while RCTs were conducted separately in children in Mongolia[83] and Boston, USA[84] with winter atopic dermatitis and in adolescent and adult Iranians.[85 86]

### Comparator group

First, due to ethical considerations, trials of harmful behaviours such as alcohol drinking and smoking focus on cessation or reduction in existing users, and do not include never smokers or never drinkers as controls, unlike MR studies, which can potentially lead to differences in effect.[23] Nevertheless, the two outcomes analysed here: hypertension for alcohol intake and lipids for smoking showed generally congruent results across study types (online supplemental dataset 5).

Second, where it would be unethical to withhold already available efficacious treatments, trials will often include another active intervention in the comparator group,[23] for example, statins in the trials of effect of PCSK9 inhibitors[97 98] on LDL-C and cardiovascular events. Such RCT design can be mimicked by factorial MR estimating the interaction of multiple exposures, as shown in matching results of the equivalent MR study.[16]

### Duration of intervention

While the magnitude of effect seen in trials with long (>3 years: weight loss to treat hypertension[99 100]/T2D,[94–96] blood pressure reduction to lower CVD risk[65–67]) and short (<6 months: alcohol intake reduction to lower blood pressure,[101–104] exercise to benefit BMD[53]/depression[62 74 78–80 105]) intervention may vary, we find both can result in directional effects consistent with MR results (figure 4), although with exceptions.[73 106]

### Triangulation of MR and RCT results

Combining RCT and MR results can offer complimentary evidence on the effectiveness of interventions. Powerful examples include congruence of positive effect of high BMI on hypertension across MR[107–110] and RCT[99 100] studies, high BMI on T2D risk in MR[89–93 111] and RCTs[94–96] and the null effect of vitamin D on various glycaemic markers in diverse populations in MR,[112] RCTs[113 114] and RCT systematic review.[115]

We also found cases, where the majority of studies pointed to one direction of effect, with one MR or RCT identified as an outlier study. In these cases, having a wide array of MR and RCT studies (ideally meta-analysed) can be helpful in establishing the likely true causal direction of effect. For instance, two MR studies,[116 117] a meta-analysis of five RCT studies[118] and two RCTs[119 120] indicate no effect of vitamin E on prostate cancer incidence with one outlier RCT[121] showing benefit of vitamin E supplementation in older smokers. Similar contrary findings were found for one RCT[122] showing beneficial effect of vitamin D on preventing depressive symptoms, as opposed to null effect in four MR studies[123–126] and two RCTs.[127 128]

On the other hand, MR analyses can show spurious disagreement with the rest of the evidence base. For instance, 2 MR papers[129 130] and a meta-analysis of 16 RCT[130] studies reveal no significant effect of vitamin D on blood pressure in the general population, with the exception of one MR study[131] that indicated a blood pressure-lowering effect of higher vitamin D status. Similarly, a range of study types: 1 MR analysis,[132] 1 RCT[133] and a meta-analysis of 27 prospective cohorts[134] (only some of them RCTs) confirm a negative impact of smoking on HDL-C levels, bar one MR study showing no significant effect.[135] A series of RCT meta-analyses[136–138] support an effect of coffee consumption (especially unfiltered) on unfavourable blood profile, although this is likely explained by diterpenes[139 140] rather than caffeine, as the latter shows evidence of cardioprotective effects.[141] However, only the recent biggest MR study[140] to date

found a significant effect of coffee consumption on LDL-C and total-C levels, unlike two previous smaller MR analyses,[142 143] which found a non-significant directionally consistent relationship.

## DISCUSSION

Our study highlighted that sparsity of data in the electronic databases seriously hampers the ability to automatically parse and compare results of MR and RCT studies. Released for the first time in 2000, ClinicalTrials.Gov is the most comprehensive resource for modern RCT (only <1000 studies, out of ~2 18 000 analysed RCTs were started before 2000). Nonetheless, we found that only 13% of all completed RCTs submitted their results to ClinicalTrials.Gov, with median trial start date in 2012. Despite 2007 legislation requiring submission of RCT results to ClinicalTrials.gov within 1 year of completion (with exceptions),[144] only 38% of eligible trials for 2008–2012 submitted their results at any time[145] which rose to 64% for 2018–2019.[146] Furthermore, 60% of studies for failed agents are reported not to be published in peer-reviewed journals,[147] and in the work presented here we found MeSH annotations were missing from the majority of complex, behavioural and dietary interventions. These factors significantly hamper efforts to systematically triangulate RCT evidence with other studies.

Next, semantic triples describing conclusions of MR and RCT studies automatically extracted from literature abstracts using rule-based methods also had low coverage, with only 36% of MR and 12% of RCT studies associated with ≥1 triples. Consequently, we instead decided to focus on a detailed qualitative investigation of a series of case studies to identify the issues associated with triangulating MR and RCT studies

Combining RCT and MR results can offer complimentary evidence on the effectiveness of interventions. Powerful examples include congruence of positive effect of high BMI on hypertension across MR[107–110] and RCT[99 100] studies, high BMI on T2D risk in MR[89–93 111] and RCTs[94–96] and the null effect of vitamin D on various glycaemic markers in diverse populations in MR,[112] RCTs[113 114] and RCT systematic review.[115] We also found cases, where the majority of studies pointed to one direction of effect, with one MR or RCT identified as an outlier study. In these cases, having a wide array of MR and RCT studies (ideally meta-analysed) can be helpful in establishing the likely true causal direction of effect.

Our analysis of genetically predicted effects of perturbation of drug target protein expression on a number of conditions with trials submitted to ClinicalTrials.Gov revealed good concordance with established therapeutics for pQTLs. However, due to the limited number of proteins (n=1002) and phenotypes (n=225, many non-diseases per se) in Zheng et al,[33 148] the comparison is necessarily very preliminary. We identify only true positive cases, as false positives and true negatives are difficult to evaluate due to sparsity of drug clinical trial results

in ClinicalTrials.Gov/literature[147] and inclusion of non-disease phenotypes in MR analysis. Anecdotally, we found no MR evidence that decreased expression of PLA2G2A leads to reduced CVD, which agrees with lack of efficacy of PLA2G2A inhibitor in clinical trials.[149–151]

The mixed reliability of eQTL instruments in predicting direction of effect on the outcome could be due to a number of factors such as less than perfect correlation between mRNA and protein levels,[152] hidden pleiotropy in single instruments used in the MR analysis (directly observed for IL2RA),[153] presence of negative feedback loop involved in the drug mechanism,[154] translation into protein isoforms with distinct biological effects[155] and differential cell-type specific drug effect.[156]

The duration of intervention varies between RCTs and MR studies, with the former spanning no more than the duration of the trial, while the latter can represent durations as long as the entire lifetime (although many exposures, such as alcohol intake, will be over a shorter time period).[7] Moreover, intervention in RCTs with long follow-up is not necessarily similarly intensive throughout its duration, or may cease altogether after some time,[74 99] that is, duration of follow-up is longer than duration of intervention in order to allow accumulation of enough events and/or confirm durability of intervention effect. Examples include lifestyle interventions, such as exercise[74] or weight loss programmes[94] like the Look AHEAD trial, with median follow-up of 9.6 years, where group and individual counselling sessions took place weekly in the first 6 months and tapered off over time.[73] That is why our analysis focused on comparing direction of effect, while ignoring magnitude of effect.[157] However, in certain cases when enough reference data are available, it is feasible to compare MR and RCT effects on the same exposure difference scale.[158]

Further impediments to direct comparison between MR and RCTs include differences in outcome definition (composite[65] vs single conditions[70]). Access to rare subpopulations with existing conditions, such as cancer patients receiving specific therapy[159] which are routinely exclusively enrolled into RCTs, can be difficult in MR due to the size of GWAS biobanks relative to N required for good power.

There are also a number of interventions and outcomes with no single phenotype which could be instrumented with GWAS variants, making MR approaches difficult, although sometimes possible with innovative MR approaches.[160] This is especially true of lifestyle interventions—such as different forms of psychological therapy, complex diet regimens[161] and fasting. Absent or limited heritability of a number of interventions and conditions, such as rehabilitation and traumatic injury makes MR approaches inaccessible.

The majority of MR studies track the onset rather than progression of disease due to availability of GWAS phenotypes[162] which are often a (binary) single measurement, as opposed to multiple quantitative outcomes frequently measured in RCTs.[163] For that reason, triangulation of

MR of onset with RCTs whose intervention is targeting progression of disease, may or may not result in agreement, as seen in our comparison of the effect of exercise on schizophrenia onset/progression (discordant) or depression (concordant) and vitamin D effect on atopic dermatitis onset/progression (discordant).

Many MR studies may be underpowered due to large sample required in indirect estimation[164] as these studies are typically studies of convenience. This bias is less common in RCTs due to preregistration of study design including power analysis,[1] uncommon in MR.[165] Null effect in MR studies may be therefore spurious and not predictive of RCTs for that reason, as seen in two smaller MR studies[142 143] out of three[140] investigating the effect of coffee intake on blood lipids, contrasting with strong clinical trial[136–138] and biochemical evidence.[139 140 166]

Furthermore, the presented literature survey used a simple heuristic of reported statistically significant evidence (p value <0.05 after multiple testing correction) to compare conclusions across MR and RCT studies, which has well-known limitations.[167 168] Inclusion of the full-spectrum of scaled point estimates along with their confidence intervals will reveal a more detailed picture in triangulation of MR and RCT evidence (online supplemental box).

Overall, we find that due to difficulty in identifying sufficient number of MR–RCT pairs matched for the same exposure and outcome, we cannot derive a numerical model to quantify reliability and importance of features of MR analysis in predicting the outcome of a future RCT. However, we make several general observations regarding usefulness of triangulation[26] of RCT with MR to guide MR studies. If an RCT shows a causal relationship between an intervention and an outcome which corresponds to the one observed in MR, it can help validate the use of these genetic variants as instruments in future MR studies. Moreover, RCTs can inform MR analyses about the plausible effect sizes and so can be useful for power calculations in MR. RCTs can help identify important interactions and subgroup effects, which can further inform MR study design. For instance, if an RCT identifies that a treatment has a stronger effect in a particular subgroup of individuals (eg, women, children), they could be analysed separately using one-sample MR.

## CONCLUSIONS

Our research highlights the challenges and benefits of triangulation of MR with RCT evidence. Future efforts, outside of the scope of this work, will focus on fully quantitative approaches towards triangulation, involving magnitude of effect size and not just its presence and direction.[25] Developers of such methods will need to be mindful of discrepancies in research hypothesis, duration and intensity of exposure, outcome measures, intervention aim, underlying population characteristics, violations of test assumptions as well as statistical power of the analysis. Furthermore, automated triangulation based on electronic databases requires intensive effort towards structured capture of both MR and RCT study results and associated meta-data, as well as annotation with shared ontologies, which is still challenging using current natural language processing methods, despite constant progress.[43 169 170]

**Author affiliations**
[1]MRC Integrative Epidemiology Unit, Bristol Medical School, University of Bristol, Bristol, UK
[2]Department of Endocrine and Metabolic Diseases, Shanghai Institute of Endocrine and Metabolic Diseases, Ruijin Hospital, Shanghai Jiao Tong University School of Medicine, Shanghai, People's Republic of China
[3]Shanghai National Clinical Research Center for Metabolic Diseases, Key Laboratory for Endocrine and Metabolic Diseases of the National Health Commission of the PR China, Shanghai Key Laboratory for Endocrine Tumor, State Key Laboratory of Medical Genomics, Ruijin Hospital, Shanghai Jiao Tong University School of Medicine, Shanghai, People's Republic of China

**Contributors** TRG conceived and supervised the study as well as acquired funding. MKS curated the data and performed the main analyses. JZ inspired the study and supplied xQTL analyses. MKS, GDS, JZ and TRG wrote the manuscript. MKS is the guarantor for this work and accepts full responsibility for the work.

**Funding** We would like to acknowledge UK Medical Research Council [mc_uu_00011/4] which provided funding for our work carried out at the University of Bristol MRC Integrative Epidemiology Unit. Our funder played no role in the design of the study, analysis and interpretation of data and in writing of the manuscript. GDS works within the MRC Integrative Epidemiology Unit at the University of Bristol, which is supported by the Medical Research Council (MC_UU_00011/1).

**Competing interests** JZ, GDS and TG receive funding from Biogen for unrelated research.

**Patient and public involvement** Patients and/or the public were not involved in the design, or conduct, or reporting, or dissemination plans of this research.

**Patient consent for publication** Not applicable.

**Ethics approval** Not applicable.

**Provenance and peer review** Not commissioned; externally peer reviewed.

**Data availability statement** Data are available in a public, open access repository. All data generated or analysed during this study are included in this published article and its supplementary information files. Code used to carry out the analysis is available on GitHub: https://github.com/marynias/mr-rct. ClinicalTrials.Gov data were accessed via AACT: https://aact.ctti-clinicaltrials.org/download and analysed data subset is featured in Supplementary datasets 1 and 2. pQTL and eQTL MR analysis results are available via EpigraphDB: https://epigraphdb.org/xqtl. PubMed database can be accessed on https://pubmed.ncbi.nlm.nih.gov/ and analysed data subset is featured in Supplementary dataset 4. SemMedDB can be accessed on https://lhncbc.nlm.nih.gov/ii/tools/SemRep_SemMedDB_SKR/SemMedDB_download.html and analysed data subset is featured in Supplementary dataset 4. Case series of MR and RCT studies with matching exposures (interventions) and outcomes (conditions) is featured in Supplementary dataset 5. Supplementary datasets 1–5 are available for download on Zenodo (https://doi.org/10.5281/zenodo.8104176).

**ORCID iDs**
Maria K Sobczyk http://orcid.org/0000-0003-0000-4100
Jie Zheng http://orcid.org/0000-0002-6623-6839
George Davey Smith http://orcid.org/0000-0002-1407-8314
Tom R Gaunt http://orcid.org/0000-0003-0924-3247

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
