## [Reviewer comments · BMJ Open]

ARTICLE DETAILS

TITLE (PROVISIONAL)	Systematic comparison of Mendelian randomization studies and randomized controlled trials using electronic databases
AUTHORS	Sobczyk, Maria; Zheng, Jie; Davey Smith, George; Gaunt, Tom

VERSION 1 – REVIEW

REVIEWER	Madeira, Catarina Chronic Diseases Research Center (CEDOC), NOVA Medical School, Universidade NOVA de Lisboa, Lisbon, Portugal
REVIEW RETURNED	21-May-2023

GENERAL COMMENTS	MR studies are based on observational data. Therefore are not interventional studies. In Figure 2 the flowchart only refers to interventional studies. It would be better to complete the diagram with the data gathered and selected about MR studies.
---

REVIEWER	Xia, Junfeng Anhui University
REVIEW RETURNED	19-Jun-2023

GENERAL COMMENTS	In this work, the authors conducted a systematic comparative analysis of Mendelian randomization (MR) and randomized controlled trials (RCT) using electronic databases. They used semi-automated mining data in the public domain, including ClinicalTrials.Gov, PubMed and EpigraphDB databases and carried out a series of 26 manual literature comparisons among 54 MR and 77 RCT publications. Their analysis highlights the challenges and benefits of triangulation of MR with RCT evidence. I found the manuscript to be overall well-written and logically structured. However, I have some concerns regarding the analysis and some conclusions. These should be addressed in detail prior to publication. 1. As far as I understand, most of the conclusion in this work are based on the case studies. So, I wonder why and how the authors selected 26 intervention-outcome case studies by manual mining of the literature in Case studies of matching MR and RCTs Section?2. I can find the codes used to carry out the analysis in this work on GitHub, however, I did not see an impressive profile README on how users can get started with the project. In addition, it suggested that the authors uploaded the raw data and the processed data from ClinicalTrials.Gov, EpigraphDB and PubMed databases on GitHub. All these would definitely make the discussion of results and conclusions more convincing.3. Since most of data sources such as ClinicalTrials.Gov study data and PubMed Data were downloaded about two years ago, I
--

	suggest the authors could update the analysis results with the recent release data. 4. It is suggested that the authors could elaborate on how researchers can apply the results of this work in their future MR analyses and thus improve the predictive power and accuracy of MR.
--	---

VERSION 1 – AUTHOR RESPONSE

Reviewer: 1

Dr. Catarina Madeira, Chronic Diseases Research Center (CEDOC), NOVA Medical School, Universidade NOVA de Lisboa, Lisbon, Portugal
Comments to the Author:

MR studies are based on observational data. Therefore are not interventional studies. In Figure 2 the flowchart only refers to interventional studies. It would be better to complete the diagram with the data gathered and selected about MR studies.

We have included Figure 2 to show our process of mining of ClinicalTrial.Gov database which provides access to meta-data and sometimes results of randomized controlled trials in a structured way. We have now updated Figure 2 with an additional panel showing the process of filtering the protein QTL MR results contained within EpigraphDB, and the same process is depicted in Figure S3 for expression QTL MR results.

Reviewer: 2

Dr. Junfeng Xia, Anhui University

Comments to the Author:

In this work, the authors conducted a systematic comparative analysis of Mendelian randomization (MR) and randomized controlled trials (RCT) using electronic databases. They used semi-automated mining data in the public domain, including ClinicalTrials.Gov, PubMed and EpigraphDB databases and carried out a series of 26 manual literature comparisons among 54 MR and 77 RCT publications. Their analysis highlights the challenges and benefits of triangulation of MR with RCT evidence. I found the manuscript to be overall well-written and logically structured. However, I have some concerns regarding the analysis and some conclusions. These should be addressed in detail prior to publication.

As far as I understand, most of the conclusion in this work are based on the case studies. So, I wonder why and how the authors selected 26 intervention-outcome case studies by manual mining of the literature in Case studies of matching MR and RCTs Section?

We included 26 intervention-outcome pairs to represent a wide array of behavioural and nutritional interventions with a diverse set of common disease (cardiometabolic, neuropsychiatric, cancer, dermatological) and disease biomarker outcomes, based on our expert knowledge of the field. Our chosen exposures correspond to the top four modifiable risk factors accounting for 39% of deaths in the USA: high alcohol intake, high body mass index (BMI), lack of exercise and smoking¹. In addition, since potentially preventative effects of nutritional factors are controversial and notoriously difficult to

¹ Mokdad, Ali H., et al. "Actual causes of death in the United States, 2000." *Jama* 291(2004): 1238-1245.

evaluate using non-randomized study designs², we also included vitamins D and E as well as coffee as an intervention.

We acknowledge that this choice is somewhat subjective but we believe it to be illustrative of the current MR and RCT literature.

We have modified the manuscript to include the justification given above in the Materials and Methods section.

2. I can find the codes used to carry out the analysis in this work on GitHub, however, I did not see an impressive profile README on how users can get started with the project.

In addition, it suggested that the authors uploaded the raw data and the processed data from ClinicalTrials.Gov, EpigraphDB and PubMed databases on GitHub. All these would definitely make the discussion of results and conclusions more convincing.

We have now updated the README file in the GitHub repository to improve the documentation of the analysis workflow: <https://github.com/marynias/mr-rct>

Our current supplementary tables and datasets contain the bulk of processed data from the database sources. We cannot release all the raw data from them on GitHub due to copyright and file size restrictions. However, this data can be easily downloaded by user from their original sources, and we include links to them in the *Data sharing statement* in the manuscript (see below). Furthermore, we include the queried subset of databases in supplementary datasets available on Zenodo: ClinicalTrials.Gov (Supplementary Dataset 1-2), SemMedDB via EpigraphDB (Supplementary Dataset 4), PubMed (Supplementary Dataset 4).

Data sharing statement

All data generated or analysed during this study are included in this published article and its supplementary information files. Code used to carry out the analysis is available on GitHub: <https://github.com/marynias/mr-rct>.

ClinicalTrials.Gov data was accessed via AACT: <https://aact.ctti-clinicaltrials.org/download> and analysed data subset is featured in Supplementary Datasets 1 & 2.

pQTL and eQTL MR analysis results are available via EpigraphDB: <https://epigraphdb.org/xqtl>

PubMed database can be accessed on <https://pubmed.ncbi.nlm.nih.gov/> and analysed data subset is featured in Supplementary Dataset 4.

SemMedDB can be accessed on https://lhncbc.nlm.nih.gov/ii/tools/SemRep_SemMedDB_SKR/SemMedDB_download.html and analysed data subset is featured in Supplementary Dataset 4.

Case series of MR and RCT studies with matching exposures (interventions) and outcomes (conditions) is featured in Supplementary Dataset 5.

Supplementary Datasets 1-5 are available for download on Zenodo: <https://doi.org/10.5281/zenodo.8104176>

² Carnegie, Rebecca, et al. "Mendelian randomisation for nutritional psychiatry." *The Lancet Psychiatry* 7(2020): 208-216.

3. Since most of data sources such as ClinicalTrials.Gov study data and PubMed Data were downloaded about two years ago, I suggest the authors could update the analysis results with the recent release data.

We have now updated the manuscript to feature the latest data available from ClinicalTrials.Gov, PubMed, EpigraphDB and SemMedDB, as of 1st July 2023 (updated values in manuscript shown in red, also updated all the figures and tables, including supplementary).

4. It is suggested that the authors could elaborate on how researchers can apply the results of this work in their future MR analyses and thus improve the predictive power and accuracy of MR.

Overall, we find that due to difficulty in identifying sufficient number of MR-RCT pairs matched for the same exposure and outcome, we cannot derive a numerical model to quantify reliability and importance of features of MR analysis in predicting the outcome of a future RCT. However, we make several general observations regarding usefulness of triangulation³ of RCT with MR to guide MR studies. If an RCT shows a causal relationship between an intervention and an outcome which corresponds to the one observed in MR, it can help validate the use of these genetic variants as instruments in future MR studies. Moreover, RCTs can inform MR analyses about the plausible effect sizes and so can be useful for power calculations in MR. RCTs can help identify important interactions and subgroup effects, which can further inform MR study design. For instance, if an RCT identifies that a treatment has a stronger effect in a particular subgroup of individuals (e.g., females, children), they could be analysed separately using one-sample MR.

We have modified the manuscript to include the justification given above in the Discussion section.

VERSION 2 – REVIEW

REVIEWER	Xia, Junfeng Anhui University
REVIEW RETURNED	01-Aug-2023
GENERAL COMMENTS	The authors have addressed my comments.

³ Lawlor, Debbie A., Kate Tilling, and George Davey Smith. "Triangulation in aetiological epidemiology." *International journal of epidemiology* 45.6 (2016): 1866-1886.